# *Pichia pastoris* and the Recombinant Human Heterodimeric Amino Acid Transporter 4F2hc-LAT1: From Clone Selection to Pure Protein

**DOI:** 10.3390/mps4030051

**Published:** 2021-07-24

**Authors:** Satish Kantipudi, Daniel Harder, Sara Bonetti, Dimitrios Fotiadis, Jean-Marc Jeckelmann

**Affiliations:** Swiss National Centre of Competence in Research (NCCR) TransCure, Institute of Biochemistry and Molecular Medicine, University of Bern, CH-3012 Bern, Switzerland; satish.kantipudi@ibmm.unibe.ch (S.K.); daniel.harder@ibmm.unibe.ch (D.H.); sara.bonetti@ibmm.unibe.ch (S.B.)

**Keywords:** 4F2hc, expression, heterodimeric amino acid transporter, LAT1, membrane protein, *Pichia pastoris*, protein purification

## Abstract

Heterodimeric amino acid transporters (HATs) are protein complexes composed of two subunits, a heavy and a light subunit belonging to the solute carrier (SLC) families SLC3 and SLC7. HATs transport amino acids and derivatives thereof across the plasma membrane. The human HAT 4F2hc-LAT1 is composed of the type-II membrane N-glycoprotein 4F2hc (SLC3A2) and the L-type amino acid transporter LAT1 (SLC7A5). 4F2hc-LAT1 is medically relevant, and its dysfunction and overexpression are associated with autism and tumor progression. Here, we provide a general applicable protocol on how to screen for the best membrane transport protein-expressing clone in terms of protein amount and function using *Pichia pastoris* as expression host. Furthermore, we describe an overexpression and purification procedure for the production of the HAT 4F2hc-LAT1. The isolated heterodimeric complex is pure, correctly assembled, stable, binds the substrate L-leucine, and is thus properly folded. Therefore, this *Pichia pastoris*-derived recombinant human 4F2hc-LAT1 sample can be used for downstream biochemical and biophysical characterizations.

## 1. Introduction

Amino acids (AAs) are essential key components for all living cells. They are not only building blocks for protein synthesis but are utilized as energy, metabolites, and precursors of signaling molecules such as neurotransmitters and peptide hormones [1]. AAs are transported across the plasma membrane by transporters [2]. In humans, there are currently more than 60 AA transporters belonging to 11 different solute carrier (SLC) families (http://slc.bioparadigms.org (accessed 22 July 2021)). Among these, heterodimeric AA transporters (HATs) are structurally exceptional: (i) they are composed of a heavy subunit from the SLC3 family (SLC3A1-2) and a light subunit from the SLC7 family (SLC7A5-A11, Slc7a12, SLC7A13, and Slc7a15) [3], which (ii) are covalently linked by a conserved disulfide bridge [3,4,5,6,7]. Whereas the heavy subunit stabilizes the heterodimeric complex and facilitates its trafficking to the plasma membrane, the light, catalytic subunit constitutes substrate transport [8,9,10]. Dysfunctional or overexpressed HATs are linked to several human pathologies, e.g., tumor cell growth, Kaposi’s sarcoma-associated herpesvirus infection, glioma invasion, aminoaciduria, or cocaine relapse [3,7,11], and were only recently reviewed with a special emphasis on brain disorders [12]. The heavy, ancillary subunit SLC3A2 (4F2hc, CD98hc) is a type II membrane N-glycoprotein, composed of an N-terminal cytoplasmic domain, followed by a single α-helical transmembrane domain (TM) and a large C-terminal ectodomain (ED) [13,14,15]. 4F2hc can associate with selected light subunits to form HATs belonging to different transport systems [3]. System L is comprised of two HATs of which its light subunits, i.e., the L-type AA transporters SLC7A5 (LAT1) and SLC7A8 (LAT2) share 48% sequence identity, are not glycosylated, and are composed of 12 TMs with cytoplasmic N- and C-termini [3,16]. The supramolecular organization of these HATs was unraveled by negative stain- and cryo-electron microscopy, i.e., 4F2hc-LAT1 [14,15,17] and 4F2hc-LAT2 [10,16,18,19,20].

4F2hc-LAT1 is expressed in several tissues as the blood–brain barrier, brain, colon, fetal liver, activated lymphocytes, ovary, placenta, spleen, testis, and tumor cells [3]. It is a Na^+^-independent obligatory exchanger that mediates substrate transport across the plasma membrane with a 1:1 stoichiometry [3,7]. Large neutral L-amino acids [21] (especially L-leucine and L-histidine [22,23,24]), L-DOPA [25], and the thyroid hormones T3 and T4 [26,27] are substrates of 4F2hc-LAT1. The ancillary protein 4F2hc modulates the substrate affinity and specificity of individual AAs in 4F2hc-LAT1 and -LAT2 [24]. For cancer cells to thrive, uptake of AAs is indispensable. 4F2hc-LAT1 shows an elevated expression level in certain types of cancer and thus provides the cancer cell with neutral and essential amino acids for regulation of the mammalian target of rapamycin (mTOR) signaling pathway and nutrition [3,23,28,29,30]. Thus, 4F2hc-LAT1 represents an important target for cancer diagnosis and treatment [31].

*Pichia pastoris* is a methylotrophic yeast that is successfully used for the overexpression of mammalian recombinant proteins [32]. *P. pastoris* has several advantages over mammalian expression systems, largely in the view of molecular biology, instrumentation requirement, laboratory handling, and cost effectiveness [33,34,35]. The system also proved to be valuable to produce membrane proteins for structural and functional studies [8,10,18,22,36,37,38,39]. In an overexpression screening campaign using *P. pastoris*, 4F2hc-LAT2 was identified to be suitable for functional and structural studies [40]. In addition, functional analyses using *Pichia* cells individually expressing 4F2hc-LAT1, 4F2hc-LAT2, LAT1, and LAT2 were reported [10,24,40,41]. This prompted us to investigate the recombinant large-scale overexpression in *P. pastoris* and purification of human 4F2hc-LAT1. Here, we describe a general applicable clone picking strategy to screen for the best transport protein expression clones. In addition to Western blot analysis, this protocol also includes protein function in the screening process. Furthermore, a procedure is given on how the human recombinant, heterodimeric, and covalently linked AA transporter 4F2hc-LAT1 is isolated from *Pichia* cells in properly folded, pure, homogenous form and good yield. The purified protein sample of this medically relevant HAT allows for further downstream biochemical and biophysical characterization, e.g., protein structure solution using novel 4F2hc-LAT1 ligands and inhibitors, or as well for the production of antibodies and other protein binders against 4F2hc-LAT1.

## 2. Materials and Methods

If not stated otherwise, all chemicals were purchased from Sigma-Aldrich Chemie GmbH Buchs, Switzerland.

### 2.1. Clone Picking Strategy

Cloning of human 4F2hc-LAT1 into the pPICZB vector (Thermo Fisher Scientific, Waltham, MA, USA) and electro-transformation using competent *P. pastoris* strain KM71H cells (Thermo Fisher Scientific, Waltham, MA, USA) was performed as described previously [24]. The construct yields recombinant human 4F2hc-LAT1 bearing N-terminal His- (4F2hc) and Strep-tags (LAT1). *P. pastoris* cells transformed with the plasmid pPICZB-4F2hc-LAT1 [24] were grown at 30 °C for 2–3 days on 0.2 mg/mL Zeocin^TM^ (InvivoGen, San Diego, CA, USA)-supplemented yeast extract-peptone-dextrose (YPD; Condalab, Madrid, Spain) agar plates. Individual colonies were repeatedly picked and streaked on Zeocin^TM^-supplemented YPD agar plates containing increasing Zeocin^TM^ concentration after each step, i.e., 1, 2, and 4 mg/mL. For each round, cells were allowed to grow at 30 °C for 2–3 days. Finally, clones, which grew at Zeocin^TM^ concentration of 4 mg/mL, were probed for 4F2hc-LAT1 expression. For that purpose, individual colonies were grown in 24 deep-well plates (EnzyScreen, Heemstede, the Netherlands), and 4F2hc-LAT1 expression levels were analyzed by Western blot analysis. In more detail, 1.25 mL of buffered glycerol-complex medium (BMGY; 1% (*w*/*v*) bacto yeast extract (BD Biosciences, Franklin Lakes, NJ, USA), 2% (*w*/*v*) peptone (Condalab), 100 mM potassium phosphate (pH 6.0), 1.34% (*w*/*v*) yeast nitrogen base (YNB; Condalab), 4 × 10^−5^% (*w*/*v*) biotin, and 1% (*v*/*v*) glycerol) were added to each well, inoculated with one colony per well, covered with a metal plate (Sandwich cover from EnzyScreen), and incubated at 30 °C and 225 rpm in an incubator (Multitron, Infors HT, Bottmingen, Switzerland) using an universal clamp (EnzyScreen). After 24 h of cell growth, protein expression was induced by diluting the cultures of each well with 1.25 mL of buffered methanol complex medium 1 (BMMY1; 1% (*w*/*v*) bacto yeast extract (BD Biosciences), 2% (*w*/*v*) peptone (Condalab), 100 mM potassium phosphate (pH 6.0), 1.34% (*w*/*v*) YNB (Condalab), 4 × 10^−5^% (*w*/*v*) biotin, and 1% (*v*/*v*) methanol). The cultures were incubated at 30 °C and 225 rpm (Multitron, Infors HT). After 24 h and 48 h of induction time, the inductive conditions were maintained by adding 250 µL BMMY5 (same as BMMY1 containing 5% (*v*/*v*) instead of 1% methanol) to each well. Then, 72 h post-induction, the cells were pelleted by centrifugation (3220× *g**,* 4 °C, 10 min). The supernatant was removed, and the cell pellets were resuspended in YPD media containing 20% (*v*/*v*) glycerol and stored at −20 °C until further usage. For SDS-PAGE sample preparation, the cell pellets were thawed, resuspended in 1 mL water, transferred into 2 mL reaction tubes (Eppendorf), and centrifuged (18,000× *g*, room temperature, 10 min). Cells were resuspended in 100 mM dithiothreitol (DTT; 100 µL per 30 mg cells) followed by incubation for 30 min at 30 °C under agitation at 1000 rpm in a Thermomixer (Eppendorf). A total of 500 µL of 20% (*w*/*v*) sodium dodecyl sulfate (SDS) per 30 mg pellet was added and further incubated for 30 min at 30 °C under agitation at 1000 rpm. Cell debris were removed by centrifugation (10,000× *g*, room temperature, 10 min), and the supernatant was transferred into a new tube, diluted 1:5 with non-reducing 5 × SDS-PAGE sample buffer (60 mM Tris-HCl (pH 6.8), 10% (*v*/*v*) glycerol, 2% (*w*/*v*) SDS, 0.01% (*w*/*v*) bromophenol blue) and separated by gel electrophoresis using a 10% SDS-PAGE gel. Overexpression levels of reduced 4F2hc-LAT1 were analyzed by Western blotting. Clones showing strong Western blot signal were used for L-leucine uptake as described previously [24]. The clone with the highest uptake signal and strong protein expression level was chosen for large-scale overexpression and purification of 4F2hc-LAT1, and thus 200 µL aliquots of glycerol stocks (at a cell density of OD_600_ = 10 in 20% glycerol) from this very clone were prepared and stored at −80 °C until further usage.

### 2.2. Large-Scale Overexpression and Membrane Isolation

For protein production, the best clone of *P. pastoris* KM71H strain expressing 4F2hc-LAT1 chosen according to procedures described in Section 2.1 was used. For this purpose, a primary seed culture of 5 mL YPD medium containing Zeocin^TM^ (1 µg/mL) was inoculated with 50 µL of glycerol stock and incubated for 24 h at 30 °C and 300 rpm (Multitron, Infors HT). From this primary seed culture, 50 µL (about 1%) was transferred into 10 × 5 mL of each fresh secondary seed culture (YPD-media containing Zeocin^TM^ (1 µg/mL)) and incubated for another 24 h at 30 °C and 300 rpm (Multitron, Infors HT) to a final OD_600_ of 6–8. The secondary precultures were combined and 6 × 8 mL were used to inoculate 6 × 800 mL of BMGY media, each supplemented with 2–3 drops of a sterile antifoaming B emulsion (Sigma). Cells were grown for 24 h at 30 °C and 300 rpm using 5 L Erlenmeyer baffled cell culture flasks, sealed with AirOtop filters (Thomson Ultra Yield, Oceanside, CA, USA) to allow for proper aeration under sterile conditions during incubation. After 24 h (at OD_600_ ≈ 50), the cells were pelleted by centrifugation (10,000× *g*, 4 °C, 10 min). Under strict sterile conditions, the cells were resuspended and induced in 800 mL of BMMY1 supplemented with 2–3 drops of a sterile antifoaming B emulsion. Cells were further incubated at 30 °C and 300 rpm (Multitron, Infors HT) using a 5 L Erlenmeyer baffled cell culture flask, which again was sealed with AirOtop filters (Thomson Ultra Yield). After 24 h and 48 h, the culture was supplemented with 1% (*v*/*v*) methanol. After 72 h, cells were harvested by centrifugation (10,000× *g*, 4 °C, 10 min). The pellet was resuspended in 50 mM sodium phosphate (pH 7.4), 10% (*v*/*v*) glycerol, 1 mM ethylenediaminetetraacetic acid (EDTA), and 50 mM β-mercaptoethanol (BME) and lysed by sonication (Brandson, Danbury, CT, USA) followed by 5 consecutive microfluidizer (Microfluidics, Westwood, MA, USA) cycles at 1500 bar. By low-speed centrifugation (10,000× *g*, 4 °C, 10 min), we separated cell debris from crude membranes, which were collected by ultracentrifugation (150,000× *g*, 4 °C, 1 h). The crude membranes were washed by homogenization in 50 mM Bis-Tris propane (BTP)-HCl (pH 8.0), 300 mM NaCl, and 10% (*v*/*v*) glycerol, and collected by a second ultracentrifugation run (150,000× *g*, 4 °C, 1 h). The washed membranes were homogenized, diluted in 50 mM BTP-HCl (pH 8.0), 300 mM NaCl, and 10% (*v*/*v*) glycerol to a concentration of 250 mg/mL, then finally flash frozen in liquid nitrogen and stored at −80 °C until further use.

### 2.3. Protein Purification

An aliquot of 500 mg membranes containing overexpressed human 4F2hc-LAT1 was thawed and solubilized in 7 mL of a buffer composed of 50 mM BTP-HCl (pH 8.0), 300 mM NaCl, 10% (*v*/*v*) glycerol, 5 mM oxidized glutathione, and 1.5% (*w*/*v*) lauryl maltose neopentyl glycol/cholesteryl hemisuccinate (LMNG/CHS, in a 5:1 ratio (w/w), Anatrace, Maumee, OH, USA) under gentle mixing for 2 h at 4 °C. The detergent solubilized fraction was separated by ultracentrifugation (100,000× *g*, 4 °C, 1 h), and the heterodimeric 4F2hc-LAT1 protein complex was purified by two sequential affinity chromatography steps. First, the supernatant was diluted 1:1 with NTA-wash buffer (100 mM BTP-HCl (pH 8.0), 300 mM NaCl, 2% (*v*/*v*) glycerol, 0.05% (*w*/*v*) glyco-diosgenin (GDN, Anatrace) and 20 mM imidazole) containing 1 mL of in NTA-wash buffer pre-equilibrated nickel-nitrilotriacetate resin (Ni-NTA; Transgenbiotech, Beijing, China) and incubated for 2 h at 4 °C under gentle agitation. The resin-containing solution was transferred into an Econo-Column (Bio-Rad), the flow-through was collected, and the resin was washed with 12 column volumes of NTA-wash buffer. The protein was eluted from the Ni-NTA resin with 100 mM BTP-HCl (pH 8.0), 300 mM NaCl, 2% (*v*/*v*) glycerol, 0.05% (*w*/*v*) GDN, and 200 mM imidazole by gravity flow in 500 µL fractions. Protein concentrations were determined at 280 nm by NanoDrop One^C^ (Thermo Fisher Scientific), and protein-containing fractions were pooled. The protein solution was desalted using Zeba^TM^ Spin columns (Thermo Fisher Scientific) and Strep-wash buffer (100 mM BTP-HCl (pH 8.0), 150 mM NaCl, 2% (*v*/*v*) glycerol, and 0.05% (*w*/*v*) GDN). Second, 2 mL of Strep-Tactin^®^XT Superflow^®^ high-capacity resin (IBA Lifesciences, Goettingen, Germany) was equilibrated with Strep-wash buffer and the 4F2hc-LAT1 sample loaded on the resin. The flow-through was collected by gravity flow and subjected again to the same resin. This procedure was repeated two times until finally four full passages of protein sample were performed. The resin was washed with three column volumes of Strep-wash-buffer and the protein was eluted from the Strep-resin with 100 mM BTP-HCl (pH 8.0), 150 mM NaCl, 2% (*v*/*v*) glycerol, 0.05% (*w*/*v*) GDN, and 50 mM D-biotin (IBA Lifesciences) by gravity flow in 500 µL fractions. Protein concentrations were determined at 280 nm by NanoDrop One^C^, and protein-containing fractions were pooled, yielding a pure sample of the heterodimeric 4F2hc-LAT1 complex for downstream applications.

### 2.4. Size-Exclusion Chromatography

A double affinity purified 4F2hc-LAT1 sample was concentrated to ≈1 mg/mL using an Amicon 100-kDa molecular weight cut-off device (Merck) followed by an additional centrifugation run (20,000× *g*, 4 °C, 5 min). Size-exclusion chromatography (SEC) was performed using a Superose 6 3.2/300 column connected to an Äkta Purifier (GE Healthcare) operated at a flow rate of 25 µL/min. After the column was equilibrated with SEC buffer (20 mM BTP-HCl (pH 8.0), 150 mM NaCl, 2% (*v*/*v*) glycerol, 0.05% (*w*/*v*) GDN), 50 µL of the concentrated protein sample was injected, eluted with SEC-buffer, and monitored by UV absorption at 280 nm.

### 2.5. Gel Electrophoresis and Western Blot Analysis

Denaturing SDS-PAGE was performed using home-made 10% SDS-PAGE gels run at constant 200 V. Precision Plus Protein Dual-color standard (Bio-Rad) was used as a molecular weight marker and protein visualization was either carried out by Coomassie Brilliant Blue R-250 (AppliChem, Darmstadt, Germany) staining or Western blot analysis. For SDS-PAGE analysis by Coomassie staining or Western blot, 1.5 µg or 0.3 µg of protein was loaded per well, respectively. For Western blot analysis, the proteins were transferred to a methanol-activated polyvinylidene difluoride membrane (PVDF; Immobilon-P Transfer Membrane, Merck Millipore) using freshly prepared transfer buffer (48 mM Tris, 39 mM glycine, 1.3 mM SDS, 20% (*v*/*v*) methanol) and a semi-dry blotting system (Trans-blot SD Semi-Dry Transfer Cell, Bio-Rad) operated at 22 V for 25 min at room temperature. The co-expressed His- and Strep-tag at the N-terminal end of either 4F2hc or LAT1 were detected using specific antibodies against these tags. Briefly, post-transfer, the PVDF membrane was rinsed with Tris-buffered saline (TBS; 10 mM Tris-HCl (pH 8.0), 150 mM NaCl), then incubated in 30 mL blocking solution (3% bovine serum albumin (BSA) in TBS) on an orbital shaker for 1 h at room temperature. After removal of the blocking solution, the membrane was directly incubated for 1 h in 30 mL blocking solution containing the appropriate antibody. For His-tag detection, Penta-His^TM^Antibody (αHis-tag-antibody, Qiagen, Germantown, MD, USA) served as primary antibody at a dilution of 1:3000 and goat anti-mouse IgG (H + L) horseradish peroxidase (HRP) conjugate (Biorad) as secondary antibody at a dilution of 1:3000. The Strep-tag was detected applying an HRP-conjugated streptavidin antibody (αStrep-tag-antibody, StrepMAB-Classic-HRP, IBA Lifesciences) at a dilution of 1:30,000. After each antibody treatment, the PVDF membrane was washed three times for 10 min with 30 mL TBS. Finally, the PVDF membrane was washed three times for 10 min with 30 mL TBS-T (TBS containing 0.05% (*v*/*v*) Tween-20), once for 10 min with 30 mL TBS followed by incubation in 3 mL electrochemiluminescence solution (Amersham ECL, GE Healthcare) for 2 min. Subsequently, antibody-labelled proteins were detected by exposing X-ray films (Fujifilm).

### 2.6. Determination of the Thermostability

In order to determine the inflection temperature (*T*_i_) of 4F2hc-LAT1 in SEC-buffer, we diluted the SEC-purified protein sample to 1 µM. Specialized glass capillaries (Nanotemper, Munich, Germany) were filled with ≈8 µL protein solution and the samples were analyzed by applying a constant heat rate of 30 °C/min from 35 to 95 °C. The fluorescence signals at 350 and 330 nm were monitored, and the 350 nm by 330 nm ratio plotted against the temperature. The *T*_i_ was elucidated by calculation of the first derivative of thermograms using a Tyco NT.6 device (NanoTemper).

### 2.7. Negative-Stain Transmission Electron Microscopy

A double affinity chromatography and SEC-purified 4F2hc-LAT1 sample solubilized in GDN at 50 µg/mL was adsorbed for 3 s on parlodion carbon-coated copper grids, which were rendered hydrophilic by glow discharge at low pressure in air. The grids were washed with three drops of double-distilled water (3 × 200 µL) and stained with two drops of freshly prepared 0.75% (*w*/*v*) uranyl formate (2 × 7 µL). Transmission electron microscopy (TEM) was performed using a FEI Tecnai F20 electron microscope operated at 200 kV and equipped with a Falcon III direct electron detector camera (Thermo Fisher Scientific). Images were recorded at a magnification of 80,000 × and a defocus of −1 µm for 2 s with an accumulated dose on the specimen level of approximately 60 e^-^/Å^2^ per exposure.

### 2.8. Scintillation-Proximity Assay (SPA) Analysis

The double affinity purified 4F2hc-LAT1 (or the purified L-arginine/agmatine exchanger AdiC; for purification procedure see [42]) was attached via the His-tag to polyvinyltolune (PVT) copper His-tag SPA beads (Perkin Elmer), and bound [^3^H]L-leucine was quantified using a scintillation counter (a detailed description of SPA is given in [43]). Experiments were conducted in 96-well plates, and, per well, a reaction volume of 50 µL contained 250 µg PVT-SPA-beads, 1.75 µg 4F2hc-LAT1 (or AdiC), 2.5 µM L-leucine spiked with 0.5 µCi [^3^H]L-leucine (ARC/Anawa, 100 Ci/mmol, 1 mCi/mL), plus a condition-specific substance. All components were dissolved in SPA buffer (100 mM BTP (pH 8), 150 mM NaCl, 10% glycerol, 0.5% (*w*/*v*) GDN). Per reaction, 5 µL of a 10 × stock solution of the condition-specific substance was placed in a well and diluted with 45 µL of a master mix containing the remaining components. To avoid minor unspecific binding of radioactive leucine to the beads, SPA-beads were first mixed with the unlabelled leucine and shaken for 2 h at 4 °C before adding the remaining components of the master mix. The plate was shortly mixed on a plate shaker and incubated at 4 °C for ≈18 h before the signals were counted using a scintillation counter (Trilux Microbeta, Perkin Elmer). Three experiments with protein from two different purifications were performed each at least in triplicate.

## 3. Results and Discussion

### 3.1. Clone Selection Procedure

A medium throughput procedure based on three individual steps was performed to select *P. pastoris* clones expressing the covalently linked heterodimeric complex 4F2hc-LAT1 at high levels (Figure 1), of which its individual subunits were either N-terminally His-tagged (4F2hc) or Strep-tagged (LAT1). First, 160 clones were picked from agar plates containing a low Zeocin^TM^ concentration of 0.2 mg/mL and streaked on agar plates. The procedure was continued by sequentially increasing the Zeocin^TM^ concentration after each round (Figure 1, Step 1) to screen for the highest copy number of 4F2hc-LAT1 integrants into the *Pichia* genome [37,44]. Second, 28 clones, which grew on agar plates containing 4 mg/mL Zeocin^TM^, were selected and used for small-scale protein expression experiments in 24-well plate format. Expression levels of 4F2hc-LAT1 were determined by SDS-PAGE under reducing conditions and immunoblotting using an αStrep-tag-antibody for protein detection; thus, the two bands visible in the Western blot displayed intact and reduced 4F2hc-LAT1 (Figure 1, Step 2). Third, clones displaying a high expression level in Western blots were selected for whole cell L-leucine uptake experiments (Figure 1, Step 3) as previously described [24] (see Figure 1, Step 3). The clone showing the highest uptake signal was finally chosen for large-scale overexpression and protein purification experiments. Importantly, clone no. 8, which in the example given showed the highest expression level in Western blots (Figure 1, Step 2, C8) was not necessarily the one with the highest uptake rate (Figure 1, Step 3, C5) and *vice versa*. Therefore, the medium throughput three-step overexpression clone selection procedure presented does not only allow for screening of clones, indicating high protein expression levels but also integrating protein function in the selection process. In addition, different constructs and expression conditions, e.g., differently tagged proteins, expression times, and methanol concentrations, may be screened individually to elucidate best expression strategy for the target membrane protein.

### 3.2. Large-Scale Expression and Purification

Large-scale overexpression runs of 4F2hc-LAT1 were performed in baffled culture flasks to ensure an efficient aeration of the culture during the cell growth and protein expression procedure. In addition, these flasks were sealed by enhanced disposable seals, which allowed in parallel a high gas exchange and the capacity to work under sterile conditions. *P. pastoris* cell walls are extremely rigid [45], and thus cell lysis poses a challenging operation, especially for large-scale preparations with the aim of membrane protein isolation. Addition of a high concentration of DTT proved useful for small-scale cell lysis but resulted in partial cleavage of the covalent linkage between 4F2hc and LAT1 (see Figure 1, Step 2). By changing the reducing agent from DTT to β-mercaptoethanol (BME) and decreasing its concentration, we were able to lyse *Pichia* cells using a combined chemical and mechanical procedure (see Section 2.2), which ultimately led to cell lysis yielding 4F2hc-LAT1 with an intact disulfide bride (Figure 2).

After cell lysis, crude membranes were isolated by three consecutive centrifugation steps involving a high salt wash (300 mM NaCl) to reduce the amount of soluble and membrane-associated contaminants that may compete with detergent extraction efficiency or unspecific protein binding to affinity resins. From 1 L of cell culture, 7-9 g of 4F2hc-LAT1-containing membranes could be isolated (Table 1), which were subjected to protein purification. On the basis of our purification protocol, we found that about 80% of the membrane mass could be efficiently solubilized using a detergent mixture composed of LMNG and CHS (Table 1). To purify detergent-solubilized 4F2hc-LAT1 from remaining contaminants and potential free 4F2hc and LAT1, we applied a double affinity chromatography purification procedure. We made use of the His-tag of 4F2hc first and bound the protein to a Ni-NTA resin. Due to the stabilizing effect of steroid-based detergents or mixtures thereof on system L HATs [18,22], and for the sake of convenience to compare results obtained from this study to previously reported ones [14,16], we exchanged the detergent mixture LMNG/CHS with glyco-diosgenin (GDN) during this step of protein purification. After protein elution from the Ni-NTA resin, partially pure 4F2hc-LAT1 could be isolated. Subsequently, the second affinity chromatography made use of the Strep-tag of LAT1 and yielded pure 4F2hc-LAT1. The double affinity chromatography purified 4F2hc-LAT1 heterodimeric complex was analyzed by SDS-PAGE and Western blotting under non-reducing and reducing conditions (Figure 2). The Coomassie Brilliant Blue-stained SDS-PAGE gel revealed a prominent band at ≈140 kDa corresponding to 4F2hc-LAT1 heterodimer (Figure 2, lane 1). In addition, negligible contributions of mainly 4F2hc monomers were discerned (Figure 2, lane 1). The latter observation of minor amounts of free 4F2hc and also possibly LAT1 might reflect SDS-based denaturation of the complex due to SDS-PAGE, which was previously observed by SDS- and blue native PAGE, and corroborated by size exclusion chromatography (SEC) of 4F2hc-LAT2 [16,19]. Western blots probed with either αHis (Figure 2, lane 3) or αStrep antibodies (Figure 2, lane 5) under non-reducing conditions confirmed the high purity and homogeneity of 4F2hc-LAT1 by also displaying prominent bands at 140 kDa in either case. Disruption of the heterodimer by breaking the disulfide bond between 4F2hc and LAT1 was performed by incubation of 4F2hc-LAT1 with DTT. Several bands corresponding to glycosylated 4F2hc at 60–70 kDa, and one band compatible with LAT1 at ≈40 kDa appeared (Figure 2, lanes 2, 4, and 6). Free LAT1 migrates faster than its expected molecular weight of 57 kDa. Such discrepancies are commonly observed for α-helical membrane proteins and have been previously reported [46,47,48]. In summary, the human heterodimeric glycosylated complex 4F2hc-LAT1 was isolated by a double affinity purification procedure with an average yield of ≈8 mg per liter *P. pastoris* cell culture (Table 1). This pure, homogenous, and structurally intact sample allows for downstream biochemical and biophysical experiments or structural studies.

### 3.3. Characterization of Purified Human 4F2hc-LAT1 Protein

To further verify the homogeneity of the GDN-solubilized 4F2hc-LAT1 heterodimer, we performed SEC. The chromatogram of 4F2hc-LAT1 displayed a prominent, sharp peak, indicating that the double affinity purification procedure applied yielded a pure and monodisperse membrane protein sample (Figure 3A). The elution volume was similar to the GDN-solubilized heterodimeric complex of 4F2hc-LAT2, as previously reported [16]. Since 4F2hc-LAT1 and 4F2hc-LAT2 share a high sequence similarity of 48% and are structurally similar [14,16,20], these data, together with the results obtained from SDS-PAGE and Western blot analysis (Figure 2) further, corroborated correct assembly of the heterodimeric complex 4F2hc-LAT1. SDS-PAGE analyses of the SEC-peak fraction by Coomassie Brilliant Blue staining showed a strong band at ≈140 kDa and faint bands between 60 and 75 kDa (Figure 3B) as prior to SEC (see Figure 2, lane 1). In addition, the SEC elution profile basically displayed no aggregation peak nor peaks resulting from cleaved 4F2hc and LAT1. These findings strengthened our hypothesis that as with 4F2hc-LAT2 [16,19], also in case of 4F2hc-LAT1, free 4F2hc and possibly also LAT1 are truly artefacts of SDS-PAGE. Furthermore, the thermostability of GDN-solubilized 4F2hc-LAT1 was measured by Tycho NT.6 analysis, and an inflection temperature (*T*_i_) of 66.3 °C was determined (Figure 3C). A study to elucidate the impact of the purification detergent on the *T*_i_ of membrane proteins was recently published [49]. Here, the authors reported on the *T*_i_ values of different α-helical membrane transport proteins solubilized in GDN to be in the range of about 50–63° C. The *T*_i_ value of our GDN-solubilized 4F2hc-LAT1 was ≈3 °C higher than these previously reported values and thus reflects heat stable transport protein sample.

Another method to assess the homogeneity and simultaneously also the low-resolution structure of protein complexes is negative-stain transmission electron microscopy (TEM). For that purpose, GDN-purified 4F2hc-LAT1 was adsorbed on parlodion carbon-coated grids, washed, negatively stained, and examined by TEM. The electron micrograph in Figure 4A shows a homogenous distribution of particles. 4F2hc-LAT1 particles, as viewed from the side, display the typical bilobed shape of 4F2hc-SLC7 heterodimeric particles [10,14,16,18,19,20,50], which are composed of a larger and smaller density (Figure 4B). The dimensions of GDN-purified, recombinant human 4F2hc-LAT1 particles are ≈12 nm (major axes) and ≈10 nm (minor axis). Compared to cryo-EM volumes of GDN-purified 4F2hc-LAT1 [14] or 4F2hc-LAT2 [16], the measured dimensions and shape of our recombinant 4F2hc-LAT1 sample purified from *P. pastoris* are in excellent agreement.

Detergent solubilization of membrane proteins may have an impact on their functional properties [49]. Thus, to determine if our detergent purified recombinant 4F2hc-LAT1 is still capable to bind substrates, we performed scintillation proximity assay (SPA) experiments. In order to validate the assay and exclude unspecific substrate binding, we carried out various experiments with known and expected outcomes. SPA experiments were performed (i) in the absence of 4F2hc-LAT1, (ii) using detergent-purified AdiC (a prokaryotic L-arginine/agmatine exchanger [51,52]), (iii) in the presence of imidazole (which will detach bound 4F2hc-LAT1 from the SPA-beads) and competition experiments using L-amino acids such as (iv) 4F2hc-LAT1 substrates L-leucine and L-histidine, and (v) the non 4F2hc-LAT1 substrate L-proline [23,24].Whereas a clear [^3^H]L-leucine binding signal could be observed in the absence of competitor (Figure 3D, black bar), all negative control experiments displayed signal reduction to background levels or no competition with L-leucine (Figure 3D, orange bars). On the other hand, the good 4F2hc-LAT1 substrates L-leucine and L-histidine strongly reduced the SPA signal (Figure 3D, blue bars). Importantly the competition experiments shown are in line with previously published functional data [23,24], i.e., L-leucine and L-histidine showed strong and L-proline no-binding competition. In conclusion, our pure, double-affinity-purified recombinant 4F2hc-LAT1 binds L-amino acid substrates in a comparable manner as previously published and is thus considered to be structurally intact.

## 4. Conclusions

We have provided a protocol that includes a protein functional assessment to conveniently screen for the most promising *P. pastoris* clones for large-scale membrane transport protein expression in a medium throughput manner. Second, an overexpression and purification procedure for the production of the recombinant human 4F2hc-LAT1 heterodimeric amino acid transporter is provided and discussed. Following this protocol, one can obtain pure, structurally intact, homogenous, thermostable, and properly folded protein in quantities that allow for downstream biochemical and biophysical characterization as structural studies or to produce antibodies and other protein binders.

## Figures and Tables

**Figure 1 mps-04-00051-f001:**
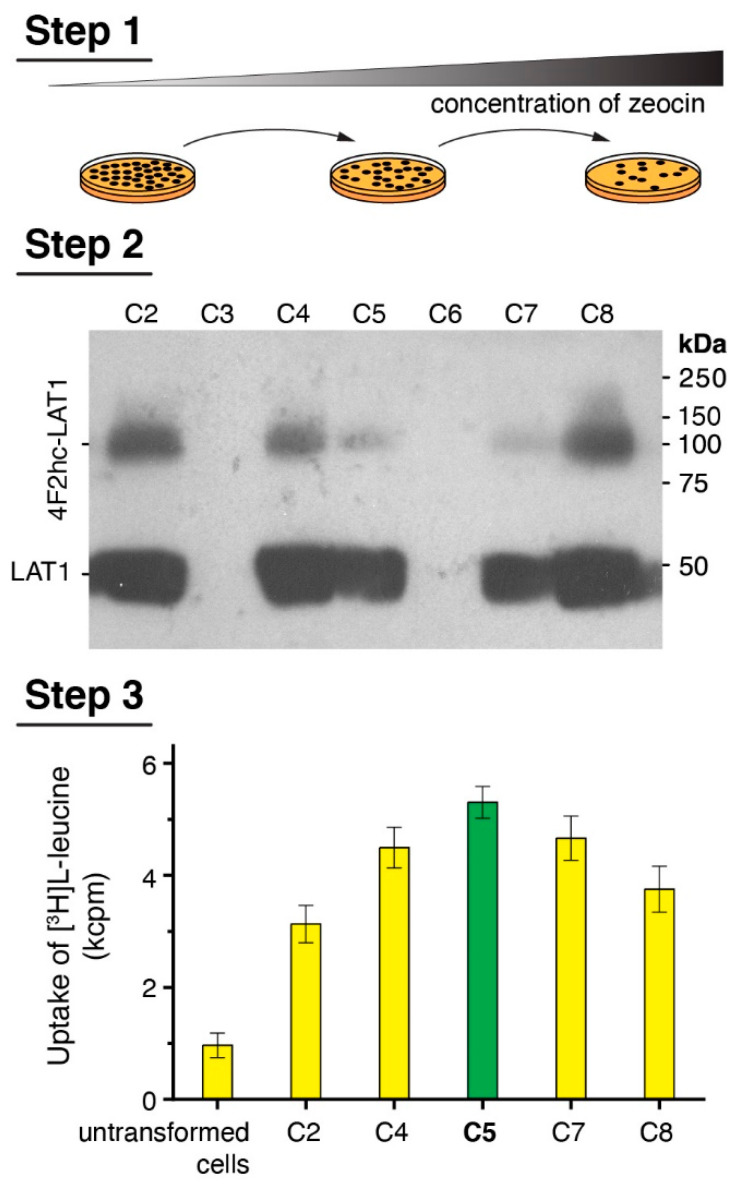
Scheme of the medium-throughput *Pichia* clone selection procedure. **Step 1** depicts a cartoon of the colony picking and streaking round by increasing the Zeocin^TM^ concentration to screen for clones with a high copy number of protein DNA sequence integrations into the *Pichia* genome. **Step 2** displays representative experimental data obtained from a small-scale 24-well plate format protein overexpression test evaluated by Western blotting. Colonies, which show relative high protein expression levels, were selected for **Step 3**. Here, Step 2 positive clones are functionally analyzed by [^3^H]L-leucine uptake into *Pichia* cells. Finally, the clone displaying highest functional property (C5, green bar) was selected for large-scale overexpression runs followed by protein purification.

**Figure 2 mps-04-00051-f002:**
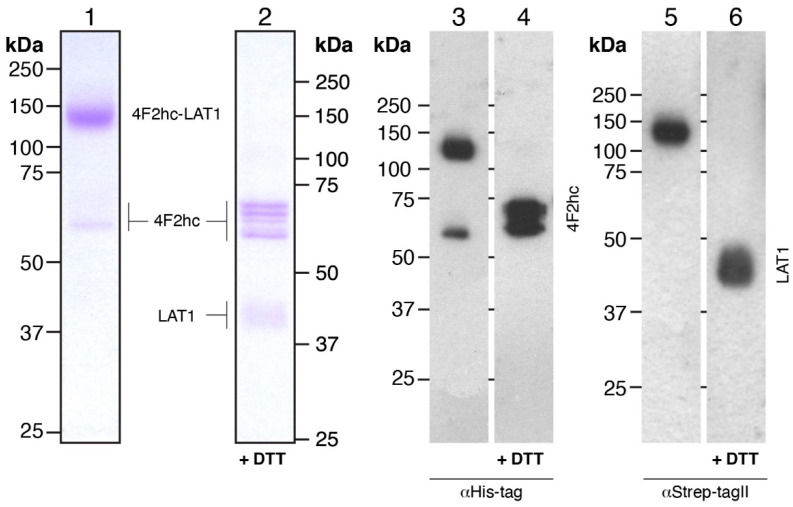
SDS-PAGE analysis of Coomassie Brilliant Blue-stained gels and Western blots. 4F2hc-LAT1 purified protein samples after double affinity purification were analyzed by 10% SDS-PAGE gels. Lanes 1 and 2 (1.5 µg of protein loaded per lane) show Coomassie-stained gels, and lanes 3-6 display Western blots (0.3 µg of protein loaded per lane). The latter were either probed with αHis-tag (lanes 3 and 4) or αStrep-TagII (lanes 5 and 6) antibodies. Lanes 1, 3, and 5 and 2, 4, and 6 depict samples run under non-reducing and reducing conditions, respectively.

**Figure 3 mps-04-00051-f003:**
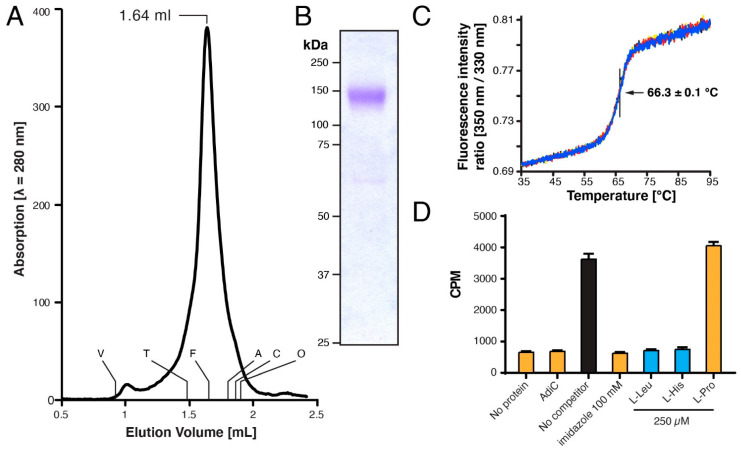
Characterization of purified human 4F2hc-LAT1 heterodimer. (**A**) Size exclusion chromatographic (SEC) elution profile of purified 4F2hc-LAT1 (≈50 μg injected) eluted with SEC buffer. A prominent monodisperse elution peak was detected at 1.64 mL. The void (V) and retention volumes are indicated for the following standard proteins: thyroglobulin (T, 669 kDa), ferritin (F, 440 kDa), aldolase (A, 158 kDa), conalbumin (C, 75 kDa), and ovalbumin (O, 43 kDa). (**B**) Coomassie Brilliant Blue-stained 10% SDS-PAGE gel of the SEC-peak fraction (1.5 µg of protein loaded). (**C**) Thermograms of the GDN-purified 4F2hc-LAT1 (1 µM) are displayed, and the *T*_i_ of 66.3 ± 0.1 °C indicated. Data are represented as mean ± SD from a quadruplicate (individual thermograms are depicted in blue, black, yellow, and red). (**D**) Scintillation proximity assay (SPA) of 4F2hc-LAT1 with the substrate [^3^H]L-leucine. Various experiments are shown, i.e., a total signal experiment (black), four negative controls (orange), and experiments using good 4F2hc-LAT1 substrates such as L-Leu and L-His (blue). Displayed is one representative experiment (of three) and counts per minute (CPM) are given with SD from technical triplicates.

**Figure 4 mps-04-00051-f004:**
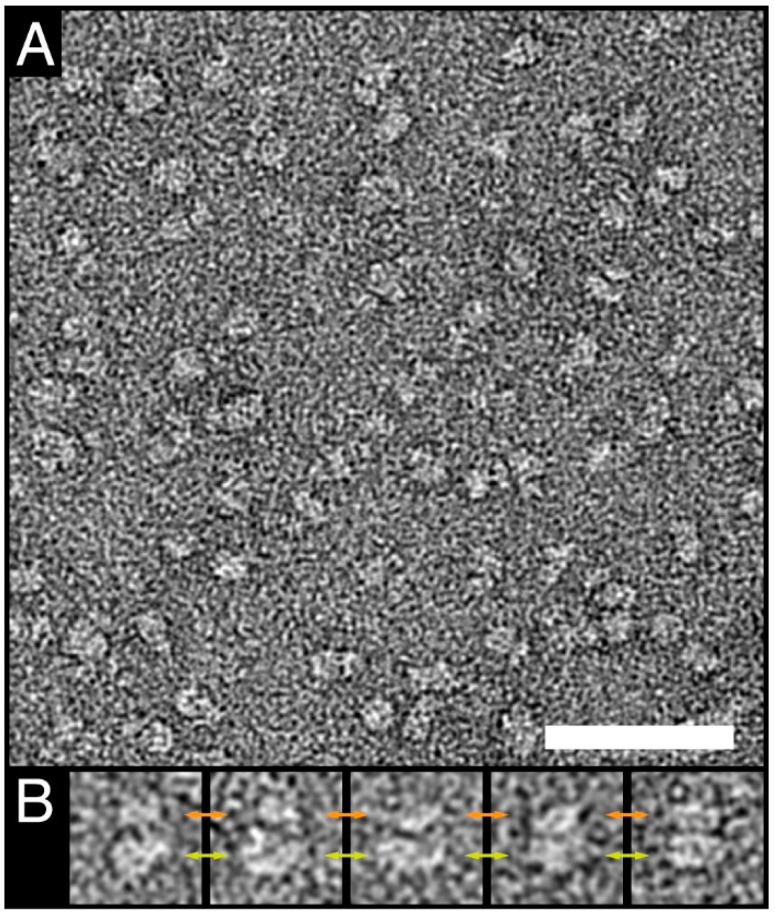
Negative-stain TEM of purified human 4F2hc-LAT1. **(A)** The electron micrograph of purified negatively stained 4F2hc-LAT1 heterodimers reflects the homogeneity of the purified protein. **(B)** Gallery of well-preserved side view particles. Typical bilobed 4F2hc-SLC7-particles composed of a larger (yellow double-headed arrows) and smaller (orange double-headed arrows) density are clearly visible. The scale bar in (**A**) represents 50 nm, and the box-size in (**B**) 20 nm.

**Table 1 mps-04-00051-t001:** Human 4F2hc-LAT1 overexpression and purification table.

Purification Procedure		Yields ^1^
Membrane preparation	Membranes isolated	7–9 g
Protein purification	Pellet after solubilization	1.0–1.5 g
	Protein amount after Ni-NTA elution	13.2–14.8 mg
	Protein amount after Strep-Tag elution	7.8–8.9 mg

^1^ Yields given are based on three independent experiments extrapolated to 1 L *P. pastoris* culture medium.

## Data Availability

Data available in a publicly accessible repository (10.6084/m9.figshare.15040143).

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
