# Peer review of "Pichia pastoris* and the Recombinant Human Heterodimeric Amino Acid Transporter 4F2hc-LAT1: From Clone Selection to Pure Protein"

_mps, 2021, doi:10.3390/mps4030051_

Round 1
Reviewer 1 Report
Introduction and Materials and methods
Line 29 and 49: across the biological membrane: please add “s” to membrane or “plasma” if you refer just to plasma membrane
Line 34: why some gene names are in bold and other name not?
Line 59: for structural studies: please add “and functional studies” (citing for example ref 8 and 22)
Line 79: 0.2 µg /ml Zeocin: in ref [24] it was reported 200 µg, please give an explanation
Line 81: what is the advantage to assay increasing zeocin concentrations in different rounds ? Why not assay directly the higher concentration as previously done (see ref [24])?
Results and Discussion
Line 223: 28 clones: in Fig 1 I find just 7 clones
Fig 1 step2: Please add the name of the proteins in corrispondence of the different bands, as done in Fig 2. In Fig 1 they should be LAT1 and the complex but with respect to Fig 2 LAT1 seems higher and the complex seems lower, please explain
Line 251: mechanical procedure…please add “see 2.2”
Lines 318-320 Fig 3D should be described in the text of the results and not in the legend. Moreover, it would be important to show the control, that is no interaction of the labeled leucine with the beads used without the attached complex. Then, I don't consider this approach a safe way to definitely demonstrate proper folding of the complex: the only method that leaves no doubt is to follow transport reconstituting purified proteins in proteoliposomes.
Finally, it is not clear the advantages given by the present procedure (concerning clone picking strategy and large overexpression) with respect to that previously pointed out (see ref. [24]). May be an improvement in the yeald of the expressed protein with respect that produced in [24]? Please write it in the text (maybe in Conclusion section)to better capture the novelty.
Please modify Conclusions according to the reviewer observations
Author Response
Dear Reviewer,
the authors wish to thank for the positive feedback to our manuscript.
The revised version of the manuscript has been improved and all points addressed accordingly as can be seen in the point-to-point discussion attached.

Reviewer 2 Report
The article by Kantipudi et al, entitled "Pichia pastoris and the recombinant human heterodimeric amino acid transporter 4F2hc-LAT1: from clone selection to pure functional protein" provides an interesting starting point for the expression and purification of the 4F2hc-LAT1 heterodimer in Pichia pastoris. The article is well written, easy to read and the protocols are clear and complete. The results are very interesting and overall I think it could be relevant for researchers working in this field. However, I have a number of comments which I will detail below.
Major points
Regarding scintillation proximity assay, I have serious doubts that it can be used as a functionality test for purified protein. On the one hand, because substrate binding as such does not necessarily imply that the purified transporter is able to translocate it, but only to bind it. On the other hand, binding needs not necessarily occur at the substrate binding site, but could take place elsewhere, and even in a non-specific manner. Thus, the apparent affinity and binding values published for HATs (Meier et al., 2001; Errasti-Murugarren et al., 2019), which report values in the mM range, do not fit the use of low micromolar concentrations for the experiment. In this sense, I propose the reconstitution of the protein in liposomes in the same way that has been carried out recently for the same protein (Lee et al., 2019; Yan et al., 2019). In this way, the functionality of the purified protein would be demonstrated.
Likewise, the SPA experiments, if they are to be kept in the article, should be improved in the following way:
- Introduce a negative control (4F2hc) binding.
- Use other LAT1 substrates (L-valine, L-histidine, L-tryptophan) as positive controls.
- Use other non-LAT1 substrate amino acids as negative controls (e.g. L-arginine, L-glutamate, L-proline).
On the other hand, the expression protocol does not appear to be optimised based on the results shown in Figure 1. In Figure 1, a large amount of LAT1 expression alone is observed, while heterodimer expression is minimal in comparison. Have different induction conditions, different constructs, etc. been tested in order to optimise expression? In my opinion this is a rather shocking aspect and at least deserves a discussion in the text.
Minor points:
Title: Functionality of the protein has not been checked, just binding, so unless transport assays performed with purified protein to check its functionality, title change must be considered.
Line 12: “4F2hc-LAT1 is a human HAT”. Change the sentence to “Human 4F2hc-LAT1” or delete “a human”.
Line 16: “Dysfunction” would not be the most suitable word to describe the role of 4F2hc-LAT1 in tumor progression. Its overexpression is associated with tumor progression while its dysfunction associates with autism (Tarlungeanu et al., 2016).
Line 37: Similarly, “Dysfunctional” would not be the most suitable word. Dysfunctional HATs are associated with aminoacidurias (b0+AT and y+LAT1), autism (LAT1) and cataracts and age-related hearing loss (LAT2), while HATs overexpression is associated with tumor progression (xCT and LAT1).
Lines 42 and 43: The sentence about the association of 4F2hc and some of the light subunits is confusing. Please rewrite it for more clarity.
Line 50: L-leucine and L-histidine are large neutral amino acids. Change the sentence “Large neutral L-amino acids as well as L-leucine and L-histidine” to “Large neutral L-amino acids such as L-leucine and L-histidine” in case you are interested in highlighting these two amino acids.
Lines 85, 94 and 107: Please homogenize units throughout the text: Indeed, mL, ml and mL appear in lines 85, 94 and 107.
Lines 128-130: Please homogenize “x g” throughout the text. In some cases there is a space between the numbers and “x g”.
Line 136: Is BTP Bis-Tris-Propane buffer? As is not a very common buffer, please define it the first time that appears in the text.
Line 162: 25 mL/min is correct? As standard flow for 3 ml columns is 0,2 ml/min I was wondering whether there was an error.
Line 205: Is SPA performed after SEC? Please specify.
Line 239: “Demonstrating a high expression level of 4F2hc-LAT1” would not be the best sentence in the context of a high LAT1 expression alone. Comparison of protein expression levels between LAT1 and heterodimer precludes the utilisation of “high expression” to refer to 4F2hc-LAT1. I would suggest to change the sentence to “Finally, the highest functional….”
Figure 1: Is the molecular weight marker the same for all the Figures? MWs are not the same in Figure 1 compared to those in Figure 2 (particularly for the 120 kDa band).
Line 331: There is an additional space in the beginning of the paragraph.
Author Response

(The authors gave the same response as above.)

Round 2
Reviewer 2 Report
I would like to congratulate the authors for the significant improvement that the experiments have brought to the article. This improved version meets all the quality standards necessary to be published in MPs.
I only found an extra punctuation mark on line 42 between the end of the sentence and the reference.
